# Abnormal Behavior Monitoring Method of *Larimichthys crocea* in Recirculating Aquaculture System Based on Computer Vision

**DOI:** 10.3390/s23052835

**Published:** 2023-03-05

**Authors:** Zhongchao Wang, Xia Zhang, Yuxiang Su, Weiye Li, Xiaolong Yin, Zhenhua Li, Yifan Ying, Jicong Wang, Jiapeng Wu, Fengjuan Miao, Keyang Zhao

**Affiliations:** 1School of Marine Engineering Equipment, Zhejiang Ocean University, Zhoushan 316022, China; 2Fishery Research Institute of Zhoushan, Zhoushan 316021, China; 3College of Communications and Electronics Engineering, Qiqihar University, Qiqihar 161006, China

**Keywords:** recirculating aquaculture system, abnormal behavior monitoring, improved YOLOX, Bytetrack

## Abstract

It is crucial to monitor the status of aquaculture objects in recirculating aquaculture systems (RASs). Due to their high density and a high degree of intensification, aquaculture objects in such systems need to be monitored for a long time period to prevent losses caused by various factors. Object detection algorithms are gradually being used in the aquaculture industry, but it is difficult to achieve good results for scenes with high density and complex environments. This paper proposes a monitoring method for *Larimichthys crocea* in a RAS, which includes the detection and tracking of abnormal behavior. The improved YOLOX-S is used to detect *Larimichthys crocea* with abnormal behavior in real time. Aiming to solve the problems of stacking, deformation, occlusion, and too-small objects in a fishpond, the object detection algorithm used is improved by modifying the CSP module, adding coordinate attention, and modifying the part of the structure of the neck. After improvement, the AP_50_ reaches 98.4% and AP_50:95_ is also 16.2% higher than the original algorithm. In terms of tracking, due to the similarity in the fish’s appearance, Bytetrack is used to track the detected objects, avoiding the ID switching caused by re-identification using appearance features. In the actual RAS environment, both MOTA and IDF1 can reach more than 95% under the premise of fully meeting real-time tracking, and the ID of the tracked *Larimichthys crocea* with abnormal behavior can be maintained stably. Our work can identify and track the abnormal behavior of fish efficiently, and this will provide data support for subsequent automatic treatment, thus avoiding loss expansion and improving the production efficiency of RASs.

## 1. Introduction

Aquatic products, as an important part of human food, have made significant contributions to global food security and personal nutrition. With the increasing demand for aquatic products, the scale and farmed species of aquaculture are constantly expanding. For some fish with high economic value or strict growing environment requirements, the land-based aquaculture model which can better control environmental variables is more suitable [1].

Recirculating aquaculture systems (RASs) are a land-based aquaculture model that can be adjusted to meet different aquaculture needs [2]. Due to the characteristics of high aquaculture density and the intensive degree of RASs, once the breeding objects are infected with bacteria and parasites or the breeding environment has become unsuitable and corresponding measures cannot be taken as soon as possible, serious losses may occur. In the early stage of the above problems, breeding objects often demonstrate some abnormal behavior characteristics; timely detection and relevant measures can greatly reduce loss. Therefore, long-term monitoring and timely detection of abnormal behavior such as turning over, stuck fishponds, and problems upstream and downstream of aquaculture objects due to environmental discomfort, bacterial diseases, parasites, and other reasons can effectively reduce losses [3].

In addition to the traditional manual method for monitoring fish behavior in aquaculture [4], there are also methods based on sensors [5,6], acoustics [7], and computer vision [8]. Due to the complex aquaculture environment, the arrangement of sensors and acoustic methods is difficult, expensive, and inefficient. However, the development of computer vision technology provides a non-invasive and efficient method for aquaculture. The three-dimensional center of gravity was obtained by setting cameras in different directions facing 40 goldfish tanks, and the behavior of the goldfish under different oxygen levels was monitored [9]. Similarly, a method using computer vision techniques to describe the behavior of farmed fish under low dissolved oxygen conditions was also proposed [10]. By using 3D vision equipment to obtain the spatial coordinate position of the cultured fish, and more scientifically to control the dissolved oxygen in the water according to the behavior description, not only can visual techniques be used to describe fish behavior, but fish can also be located and tracked. In addition to directly setting the camera to observe the fish, they can describe the behavior of the fish as they pass through setting up a fish pass, combining computer vision and artificial neural network techniques to determine the position of the fish and to calculate the speed and acceleration of the fish from consecutive images [11]. By combining background subtraction, a Kalman filter, and a Hungarian algorithm, the fish in the video can be tracked and counted to achieve the purpose of monitoring [12].

In recent years, compared with traditional image processing methods, computer vision technology based on deep learning can better extract image features to complete more complex tasks. This technology brings more possibilities for scientific applications in agriculture [13,14], including aquaculture. The lightweight network mobilenetv1 [15] based on depth-wise convolution was used to replace the original backbone network to improve the YOLOV3 [16] algorithm, which greatly reduced the computational load while guaranteeing the quality of feature extraction and improving the detection accuracy of the fish [17]. In addition to the improvement in the network’s structure, a method combining multi-scale Retinex and YOLOV3 was proposed to detect the fish in the tank. The optical flow method was used to track the fish according to the detection results [18]. Subsequently, an improved YOLOV4 algorithm was proposed [19]. The improvements include reducing the number of layers of the feature extraction network, using DenseNet [20] instead of ResNet [21], and modifying neck connections. The algorithm was used to detect underwater uneaten feed pellets for better scientific farming. The single object tracking algorithm SiamRPN++ [22] has been adopted many times to successfully track the abnormal behavior of fish in fishponds in order to better promote scientific breeding [23]. However, it needs to be repeated several times to complete the tracking of multiple fish in the video.

Although computer vision methods based on deep learning have been gradually applied to aquaculture, they are still relatively few in number and are immature compared with other fields. Due to the high density of farming and the complex environment in RASs, monitoring is often difficult due to the following problems: firstly, the fish with low vitality will be stacked together, resulting in difficult detection; secondly, the activities of lower fish are often accompanied by occlusion and cover from the shadows of upper fish, which cause feature loss and missed detection; thirdly, the ripples caused by the oxygen supply equipment and the activity of the fish cause the deformation of the fish’s bodies during detection; finally, during tracking, the constant occlusion of the fish due to their activity leads to ID switching, and switching between the normal state and the abnormal state will also cause their ID to be lost.

According to the above literature and problems, this paper takes *Larimichthys crocea* in a RAS as the research object and proposes a method to detect and track the abnormal behavior of *Larimichthys crocea* in a RAS high-density scene. The method proposed in this paper includes two parts: detection and tracking. Aiming to solve the problems of *Larimichthys crocea* in RASs often accompanied by stacking, deformation, occlusion, and small objects, the above problems can be solved by the improved YOLOX [24] object detection algorithm, and the detection results are sent to Bytetrack [25] to perform data matching and complete the tracking. The method can complete the detection and tracking at the beginning of the abnormal behavior of the *Larimichthys crocea* in the RAS, which can better improve the quality of aquaculture and reduce loss.

## 2. Materials and Methods

### 2.1. Image Acquisition

The abnormal behavior dataset of *Larimichthys crocea* in the RAS used in this paper was collected between April 2022 and July 2022 at the Zhoushan Fisheries Research Institute, Zhejiang Province, China. Figure 1 shows a picture of the workshop of the RAS of the experimental materials obtained. There are six *Larimichthys crocea* ponds in the workshop. The diameter of the pond is about 6 m. The number of *Larimichthys crocea* in each pond is about 400~600. The abnormal behavior dataset collected mainly consists of the turning-over behavior of *Larimichthys crocea* caused by environmental discomfort, bacterial diseases, and parasites. The dataset acquisition devices used in the experiments are a laptop and an Intel D455 depth camera. In order to ensure the vision system has wide adaptability in the unstructured environment, we collected 52 videos with different growth periods under different lighting conditions. The obtained video resolution is 1280 × 720 and the frame rate is 30 FPS. To better train the images, a picture is captured from the video every 10 frames, with a total of 7279 photos.

### 2.2. Abnormal Behavior Detection

#### 2.2.1. YOLOX

YOLOX is a continuation of the YOLO [26] series of algorithms. It skillfully combines recent developments in the field of object detection. While ensuring efficient reasoning, its performance has also been significantly improved. According to the width and depth of the network structure, YOLOX is divided into standard networks (YOLOX-S, YOLOX-L, YOLOX-M, and YOLOX-X), lightweight networks (YOLOX-Nano and YOLOX-Tiny), and a darknet53 version. In order to meet the real-time requirements, YOLOX-S, which has the simplest network structure and the fastest detection speed, is used to detect the abnormal behavior of *Larimichthys crocea* in RASs. Its network structure is shown in Figure 2, which is mainly divided into three parts: the backbone, neck, and head.

When a picture is input, it first goes through the backbone part including the focus module and four CSP modules for feature extraction. The size of the input picture is (640 × 640 × 3), and the output size is (320 × 320 × 12) after entering the focus module for the slicing operation; then, feature extraction is performed through 4 CSP modules, and 3 feature maps are output, with the sizes of (80 × 80 × 256), (40 × 40 × 512), and (20 × 20 × 1024). After feature extraction is completed, the PAnet of the neck part performs top-down and bottom-up feature fusion on the three feature maps output by the backbone and outputs three enhanced feature maps. When outputting, the decoupling head operation is performed first. Although this makes the calculation difficult, it also improves the detection performance and convergence speed. Secondly, the network uses an anchor-free operation for better detection by reducing the parameters. Finally, SimOTA is used to dynamically allocate the number of positive samples.

#### 2.2.2. The Proposed Algorithm

Considering the complexity of the RAS environment, this paper proposes the following modifications to improve the detection accuracy for false detections and missed detections caused by stacking, deformation, occlusion, small objects, and other factors: modify the CSP structure, add coordination attention [27], and name it CA-CSP.

In the process of feature extraction, the fish stack caused by high density and the deformation caused by fish activities will lead to deviation of the extracted features, resulting in the final features being missed and false detection. The main function of the attention mechanism is to make the neural network pay attention to the required information, reduce the attention of other information, and even filter unimportant information. At present, there are many types of attention mechanisms, such as the squeeze-and-excitation (SE) network [28], the convolutional block attention module (CBAM) [29], etc. Some of them are effective in feature extraction. Some combine different features to improve the accuracy of feature extraction. Therefore, this paper improves the feature extraction module and adds coordination attention.

For the feature map with the input size (C, W, H), pooling kernels of (W, 1) and (H, 1) are applied to each channel along the X direction and Y direction, respectively. Then, the output of the C-th channel with height h is $z_{c}^{h}$, and the output of the C-th channel with width w is $z_{c}^{w}$, thus acquiring feature perception maps in two directions.

Concatenate z^h^ and z^w^, and then the results are sent to the shared 1×1 convolution transformation function F1 to generate the intermediate feature map f. The obtained intermediate feature map f is extended in the X and Y directions to obtain the directional feature map g^h^ and g^w^ again, and the final output value is shown in Formula (4).
(1)$$Z_{c}^{h}\left(h\right)=\frac{1}{W}\underset{0\leq i\leq W}{\sum }x_{c}\left(h,i\right)$$
(2)$$Z_{c}^{w}\left(h\right)=\frac{1}{H}\underset{0\leq j\leq H}{\sum }x_{c}\left(j,w\right)$$
(3)$$f=\delta \left(F1\left(\left[Z^{h},Z^{w}\right]\right)\right)$$
(4)$$y_{c}\left(i,j\right)=x_{c}\left(i,j\right)\times g_{c}^{h}\left(i\right)\times g_{c}^{w}\left(j\right)$$

The structure of the modified feature extraction module CA-CSP is shown in Figure 3. When extracting features, this modification cannot only capture cross-channel information but also retain the input information along the X and Y directions to improve the representation ability of the feature map, which can more accurately locate and identify the object of interest.

Then, add the feature map output by the backbone and modify the neck part of the network connection. In the original backbone network, the features of pictures are extracted through 4 CSP modules, but only 3 feature maps with sizes of (80 × 80 × 256), (40 × 40 × 512), and (20 × 20 × 1024) are output. The feature map with the size of (160 × 160 × 128) is lost, so many features of small objects and incomplete objects are lost during feature fusion. Therefore, this paper modifies the original network structure, including adding the feature map output of the backbone and modifying the structure of the neck part. In addition, the CA-CSP module is used to replace the CSP module in the original neck part (Figure 4a). The modified structure is shown in Figure 4b. The four feature maps are fused in the neck part and four enhanced feature maps are output to achieve the purpose of the accurate detection of small objects and occluded objects.

### 2.3. Abnormal Behavior Tracking

Different from the current mainstream trackers, Bytetrack is a tracking algorithm that is not based on deep learning nor does it use re-identification technology. The core of the algorithm is similar to Sort [30], which has a Kalman filter and a Hungarian algorithm.

Figure 5 is the tracking process of Bytetrack, which mainly includes two matches. When the video sequence is input, all of the detection boxes are firstly divided into high-scoring detection boxes and low-scoring detection boxes according to the threshold, and the trajectory is created. Secondly, Kalman filtering is used to predict the position and size of the detection boxes in the next frame, and the IOU (intersection over union: overlapping rate of the bounding box and ground truth) between the predicted detection boxes and the current high-scoring detection frame is calculated. Third, according to the obtained IOU, the created trajectory is first matched with the high-scoring detection boxes of the current frame using the Hungarian algorithm. Finally, the trajectory of the detection boxes that have been successfully matched are updated. After the first matching is completed, the trajectories that are not successfully matched in the first matching process are matched with the low-scoring detection boxes of the current frame. New tracks are created for high-scoring boxes that are not successfully matched in the second match, and low-scoring boxes are deleted.

## 3. Results

This study used the improved YOLOX-S with Bytetrack to detect and track the abnormal behavior of *Larimichthys crocea* in a RAS. A desktop computer equipped with a Win10 operating system with an 11th Gen Intel Core i9-11900k CPU and NVIDIA Geforce RTX 3080 Ti (12 g) GPU was used to train the *Larimichthys crocea* abnormal behavior dataset. The dataset is divided into the following situations: 5986 images in the training set, 728 images in the test set, and 521 images in the validation set.

### 3.1. Detection Results

This experiment uses YOLO-V4 [31], YOLO-V5 [32], and improved YOLOX to detect the abnormal behavior of *Larimichthys crocea* in a RAS and to compare their detection performance. The abnormal behavior dataset of *Larimichthys crocea* in the RAS is trained for 600 epochs with a batch size of 8, an initial learning rate of 0.01, a momentum of 0.937, and a decay of 0.0005.

The results of object detection can be divided into four types: TP (abnormal behavior is correctly detected), FP (other behavior is falsely detected as abnormal behavior), TN (other behavior is correctly detected), and FN (abnormal behavior is falsely detected as other behavior). According to the above four indicators, the different calculation methods constitute the recall and precision indicators. The curve formed by the two indexes’ recall and precision under different confidence levels is called the PR curve, and the area enclosed by this curve is the average precision (AP). Therefore, the formula for calculating AP is the integral of P (R) from 0 to 1 confidence. AP_50:95_ represents the mean value of AP with different confidence levels. The most important evaluation indicators for object detection are calculated from the above four results, as follows:(5)$$\mathrm{Recall}=\mathrm{TP}/\left(\mathrm{TP}+\mathrm{FN}\right)$$
(6)$$\mathrm{Precision}=\mathrm{TP}/\left(\mathrm{TP}+\mathrm{FP}\right)$$
(7)$$F1=\left(2\times \mathrm{Recall}\times \mathrm{Precision}\right)/\left(\mathrm{Recall}+\mathrm{Precision}\right)\;$$
(8)$$\mathrm{AP}=\int_{0}^{1}P\left(R\right)\mathrm{dR}$$
(9)$${\mathrm{AP}}_{50:95}=\left({\mathrm{AP}}_{50}+{\mathrm{AP}}_{55}+\ldots +{\mathrm{AP}}_{90}+{\mathrm{AP}}_{95}\right)/10$$

This paper uses the above indicators to evaluate the improved YOLOX-S object detection algorithm; Table 1 shows the evaluation results of the abnormal behavior detection performance of a RAS using different detectors. Compared with the original YOLOX-S, the detection performance is significantly improved. The meaning of recall is the proportion of correctly detected abnormal-behaving fish in the total number of abnormal-behaving fish in the sample. Compared with the YOLOX-S, the recall of the improved YOLOX-S increases from 89.76% to 97.95%. Precision represents the proportion of correctly detected abnormal-behaving fish in all of the detected abnormal-behaving fish, and the precision of the improved YOLOX-S reaches 97.17%. The PR curves are shown in Figure 6; the closer the curve is to the upper right, the better the performance of the detector and the area wrapped in the curve is the AP value. It is clear that the PR curve of the improved YOLOX-S algorithm proposed in the paper wraps the curves of the other detectors, which proves that the proposed improvement is effective and reasonable. F1, which integrates recall and precision, also increases from 0.91 to 0.98. Although AP_50_ only increases by 5.1% compared with YOLOX, AP_50:95_ increases from 54.4% to 70.6%, which is a significant increase of 16.2%. This not only indicates that the detection score for abnormally behaving *Larimichthys crocea* has increased, but it also means that it is more accurate for stacked objects, deformed objects, occluded objects, and small objects. Regarding the detection speed of the detector, due to the improvement of the detection algorithm in this paper, the network weight and parameters are increased to reduce the running speed. Although the frame rate is only 50.67 FPS, it can fully meet the requirements of real-time detection with higher accuracy.

### 3.2. Tracking Results

This study used Bytetrack to track *Larimichthys crocea* with abnormal behavior in a RAS. At the same time, Deepsort [33], the mainstream tracking algorithm that uses appearance feature matching to maintain ID, was used for demonstration and comparison in this paper.

When tracking abnormal behavior, the main parameters of Bytetrack are set as follows: the detection threshold is 0.4, the tracking threshold is 0.6, the tracking buffer is 300, and the matching threshold is 0.8. Since Deepsort is a deep-learning-based multi-target tracking algorithm, additional training is required for this. The abnormal behavior dataset of *Larimichthys crocea* in the RAS is trained for 40 epochs with a batch size of 64, an initial learning rate of 0.01, a momentum of 0.9, and a decay of 0.0005. The main parameters of Deepsort are set as follows: the detection threshold is 0.4, N_INIT (the number of frames to keep track) is 6, and the NN_BUDGET (save the number of successfully matched features) is 100.

The main indicators for tracking evaluation are as follows:(10)$$\mathrm{MOTA}=1-\sum_{t}\left({\mathrm{FN}}_{t}+{\mathrm{FP}}_{t}+{\mathrm{IDSW}}_{t}\right)/\sum_{t}\left({\mathrm{GT}}_{t}\right)$$
(11)$$\mathrm{IDF}1=2\mathrm{IDTP}/\left(2\mathrm{IDTP}+\mathrm{IDFP}+\mathrm{IDFN}\right)$$

In the above multiple objects tracking accuracy (MOTA) equation, FN_t_ is missed detection in the t frame, FP_t_ is false detection in the t frame, IDSW_t_ is the number of ID switches of tracking objects in the t frame, and GT_t_ is the total number of tracking objects in the t frame. IDF1 represents the ratio of correct recognition detection to the average true number and the calculated detection number. IDTP and IDFP represent the true positive ID number and the false positive ID number, respectively, and IDFN is the false negative ID number.

In Appendix A, different trackers are shown respectively to track the abnormal behavior of *Larimichthys crocea* in different complexity scenes in the actual RAS. Among them, Appendix A show the tracking results of *Larimichthys crocea* with abnormal behavior in complex scenes with more interference by using Bytetrack and Deepsort; Appendix A show the tracking performance in simple scenes, and Appendix A show the tracking results with less interference. Correspondingly, the performance evaluation of the different trackers for the tracking of *Larimichthys crocea* with abnormal behavior in different scenarios is shown in Table 2.

In complex scenarios with more interference such as in Appendix A, the MOTA of Bytetrack is 21.99% higher than that of Deepsort, with it reaching 95.02%. In terms of IDF1, Bytetrack’s performance is outstanding, with it reaching 97.7%, which is 35.9% higher than the results of Deepsort. Throughout the tracking process, the number of ID switches using Byetrack is only 1, which is far less than the number of ID switches using Deepsort. The tracker needs to run with the detector at the same time, and the FPS using the Bytetrack algorithm reaches 39.4, which can fully meet the requirements of real-time tracking. In other scenes with less interference, such as Appendix A, the difference between the MOTA values of Deepsort and Bytetrack is small, but the difference between the IDF1 values is large, which indicates that the ID of Deepsort that uses appearance information for matching to complete tracking cannot be maintained stably during tracking. Furthermore, in sequence 2 and sequence 3, the MOTA of Bytetrack reached more than 99% and the more important IDF1 also reached 96.41% and 99.53%, respectively, and no ID switching occurred during the tracking process. In general, the requirement of real-time monitoring is to process 30 frames per second. When using improved YOLOX-S combined with Bytetrack to verify video sequences, the average processing speed can reach about 39 frames per second. From the data in the table, whether it is the accuracy of tracking or the stability of the ID during the tracking process, Bytetrack, which avoids the re-identification of appearance features, is more suitable for the tracking of *Larimichthys crocea*.

## 4. Discussion

### 4.1. Detection Performance Comparison

Figure 7 shows the comparison of detection performance after adding different attention mechanisms. In the heat map, the closer the area of the red part is to the area of the detected object, the more accurate the detection is. The darker the color of the red part, the higher the detection reliability. The figure shows the detection heat map results of the original detector, the detector with CBAM added, the detector with SE added, and the detector with CA added. The images on the left show that there are many detection objects. There is no obvious missing or false detection in the detection results of each detector. However, compared with the CA added in this paper, the detected area is larger, more closely fitting to the *Larimichthys crocea* with abnormal behavior, and the color is relatively dark. The images on the right show the detection of *Larimichthys crocea* with abnormal behavior under the influence of oxygen supply equipment. The operation of the oxygen supply equipment leads to more bubbles and ripples in the pool, which seriously affects detection. Compared with the channel attention mechanism of SE and the spatial and channel combined attention mechanism of CBAM, the network with CA can obtain the spatial location information of dense objects in complex scenes more accurately by obtaining the location information of objects in the X and Y directions, so as to be more sensitive to stacked objects, covered objects, and deformed objects. Therefore, the objects missed by other detectors can be completely detected by using YOLOX-S with CA.

Figure 8 shows the detection performance of different detectors for abnormal behavior in complex scenes, which more intuitively shows the growth of the evaluation indicators. Figure 8a shows a comparison of the performance of different detectors in the detection of stacked objects. The pictures mainly show the detection results of five groups of stacked *Larimichthys croceas*. YOLOV4, YOLOV5-S, and YOLOX-S can complete the detection of the fish that are far away, while for the fish that are located closely to one another, the detector only detects one fish. However, the improved YOLOX-S completed accurate detection and frame selection for the five groups of stacked fish in the figure, thus showing excellent performance. 

Figure 8b shows that the detector detects the *Larimichthys crocea* near the oxygen supply equipment. Under the same datasets and the same training parameters, each detector can complete detection of turned-over fish. When the *Larimichthys crocea* are close to the running oxygen supply equipment, this will cause visual deformation, and only the improved YOLOX-S can accurately detect the visually deformed fish.

In the third group of the comparison pictures, the detection of *Larimichthys crocea* in the occluded state is mainly demonstrated. In the picture, a turned-over fish is occluded by two normal fish. When the picture is input into the network, due to excessive occlusions and fewer features, the occlusion object is missed. The improved YOLOX-S solves this problem successfully. 

In addition to the above problems, the improved YOLOX-S is also very accurate for small object detection. In Figure 8d, the five *Larimichthys crocea* in the target area are very small for the whole picture, and the influence of non-structural light brings great challenges to the detector. However, compared with the other detectors, the improved YOLOX-S successfully detects the five *Larimichthys crocea* with abnormal behavior in the picture.

In the actual RAS scene, due to the characteristics of high density and the interference of equipment and unstructured light, it is often impossible to achieve effective detection. In this paper, by modifying the CSP module, adding CA can make feature extraction more effective, and increasing the output number of feature maps can better fuse and strengthen features on this basis, and then it can effectively solve the problems of stacking, deformation, occlusion, small object detection, and so on.

### 4.2. Tracking Performance Comparison

Figure 9 is a comparison of tracking the same video sequence using Bytetrack and Deepsort, and the picture shown in Figure 9 is the key frame of Appendix A. Figure 9a shows frame five of the two trackers. The IDs of the two trackers in the initial state are normal and there is no interference. Due to them being occluded by passing fish, the *Larimichthys crocea* whose ID is 9 at frame 5 has switched to ID 34 at frame 130. Due to the re-identification of appearance features, the ID switched back to frame 9 at frame 150. At the same time, it can be seen that the *Larimichthys crocea* with ID 37 in the A area has two ID boxes. This is also due to the use of appearance feature matching, resulting in multiple IDs for one fish. At 260 frames, the original ID 9 of the *Larimichthys crocea* in the A area switched to ID 64, and the original ID of the *Larimichthys crocea* with the original ID 10 has been switched to ID 60.

The *Larimichthys crocea* with abnormal behavior in area B is occluded by multiple fish from frame 5 to frame 260. Among them, the ID switching times of the two *Larimichthys crocea* stacked together in the lower left of area B are greater. At frame 5, the *Larimichthys crocea* with ID 5 and ID 11 switched to ID 33 and ID 35 at frame 130, respectively. It is worth noting that the box with ID 35 is located on the *Larimichthys crocea* on the left side of the lower left corner in the B area at frame 150 but it switched to the *Larimichthys crocea* on the right side at frame 260. This is caused by the similarity of the appearance features of the fish, which is caused by the tracker using appearance feature re-identification when two *Larimichthys crocea* are stacked together.

However, Bytetrack, which does not use appearance features for matching, performs well in the tracking process. Although only the result of the detector is used to match the data, the ID is maintained when occlusion and deformation occur. In addition, there will not be a case where there are multiple IDs on one fish or the ID jumps to nearby fish.

### 4.3. Generalization Verification

To verify the generalization and robustness of the proposed method, we collected *Larimichthys crocea* at different growth stages from different aquaculture ponds at the Zhoushan Fisheries Research Institute for experimental verification. These videos were shot at an unfixed angle, and there were operations such as moving and zooming in the process of video shooting. In addition, because of the different collecting times, different fish ponds had different-colored water and different lighting conditions. The detection and tracking results are presented in Figure 10. From the start frame to the end frame, in the face of different environments and unfixed shooting, our method can still achieve better detection and stable tracking of *Larimichthys crocea* with abnormal behavior. Although there is occasional loss of ID in the process, the correct ID of each fish is guaranteed after re-matching through the cached data, which proves that our method is suitable for complex environments and unstable shooting conditions. The relevant video sequence has been placed in the Appendix A.

## 5. Conclusions

This paper presents a method for detecting and tracking the abnormal behavior of *Larimichthys crocea* in RASs. In terms of detection, an improved algorithm is proposed to solve the problems of stacking, occlusion, deformation, and small objects caused by dense objects in the fish pond and the complex environment. For detection performance, compared with the original algorithm, the AP_50_ of the improved YOLOX-S increased by 5.1% to 98.4% and the AP_50:95_ increased significantly by 16.2%. It has been proven that the improved YOLOX-S can effectively solve the detection difficulties caused by the above problems. For object tracking, Bytetrack is used to track the detected objects to avoid the problem of similar appearance features of fish and it uses the boxes obtained from detection to match to ensure the stability of tracking. In complex scenes with more interference, the MOTA and IDF1 of Bytetrack using only detection information can reach more than 95%. It can also maintain high frame rate operation, which can fully meet the effect of real-time monitoring. Our method is also suitable for a variety of complex aquaculture environments and different shooting conditions. It can complete the monitoring of the abnormal behavior characteristics of different breeding objects.

## Figures and Tables

**Figure 1 sensors-23-02835-f001:**
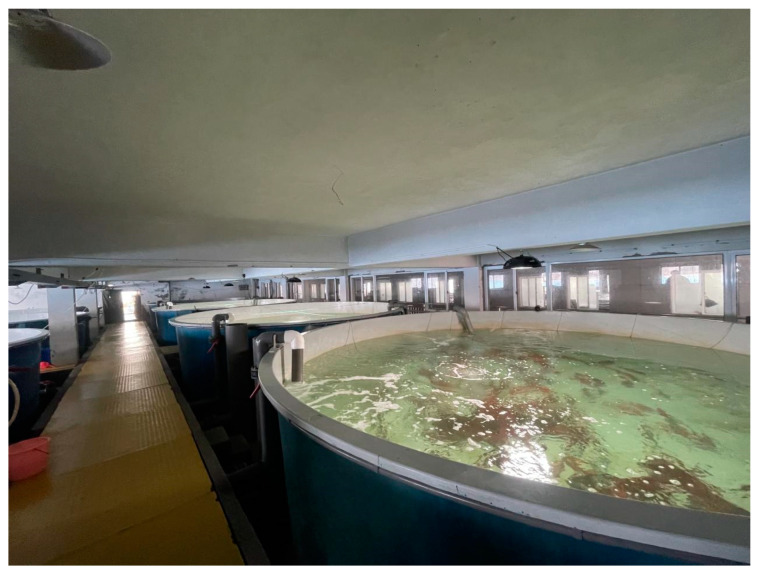
Workshop photo of the *Larimichthys crocea* recirculating aquaculture system taken at the Zhoushan Fisheries Research Institute in June 2022.

**Figure 2 sensors-23-02835-f002:**
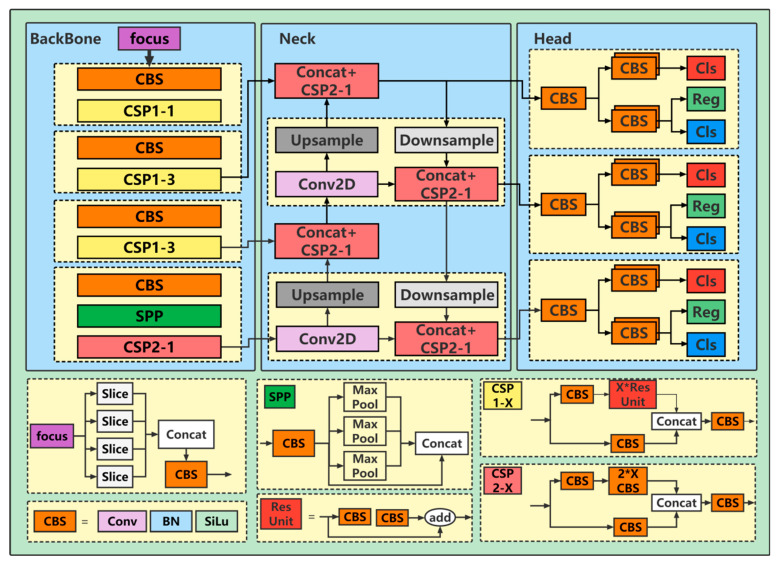
The YOLOX-S framework schematic diagram and flow, including the description of its constituent modules.

**Figure 3 sensors-23-02835-f003:**
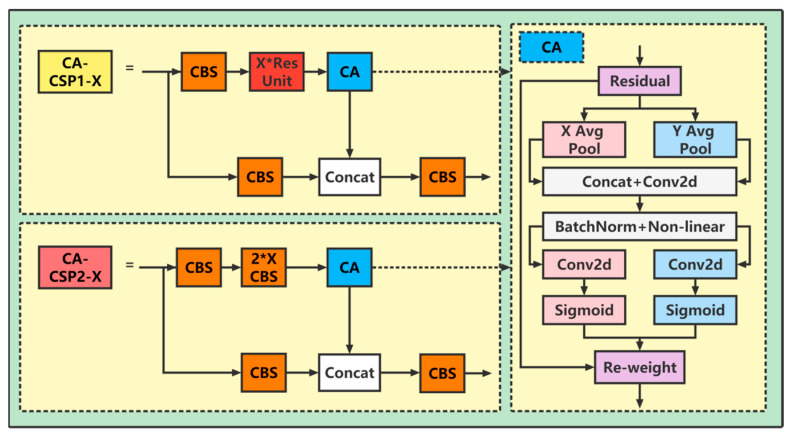
Coordinate attention module structure and CSP module structure with the coordinate attention module added.

**Figure 4 sensors-23-02835-f004:**
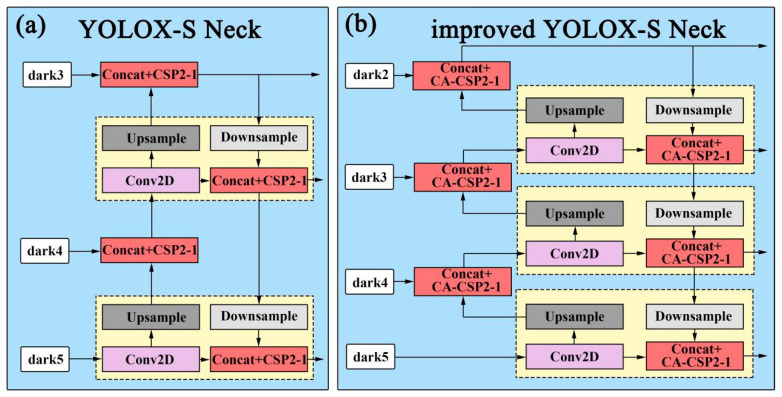
Neck part structure comparison: (**a**) the original neck structure; (**b**) the neck structure using the CA-CSP module and modifying the connection.

**Figure 5 sensors-23-02835-f005:**
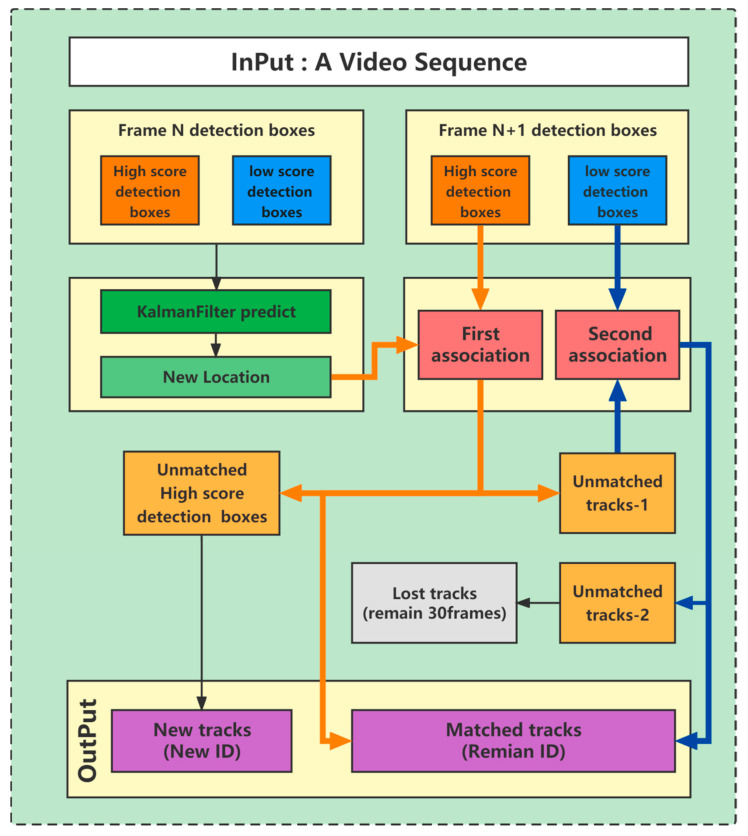
Process diagram of Bytetrack, including the tracking process, data association, and ID assignment rules.

**Figure 6 sensors-23-02835-f006:**
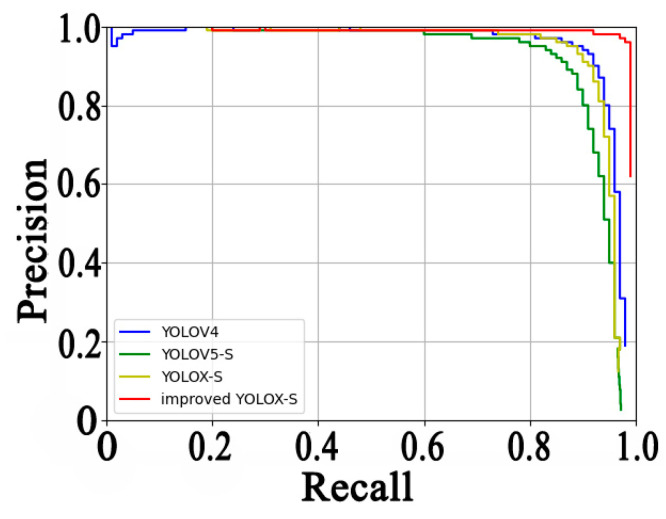
The PR curves of different detectors detecting abnormal behavior datasets of *Larimichthys crocea*.

**Figure 7 sensors-23-02835-f007:**
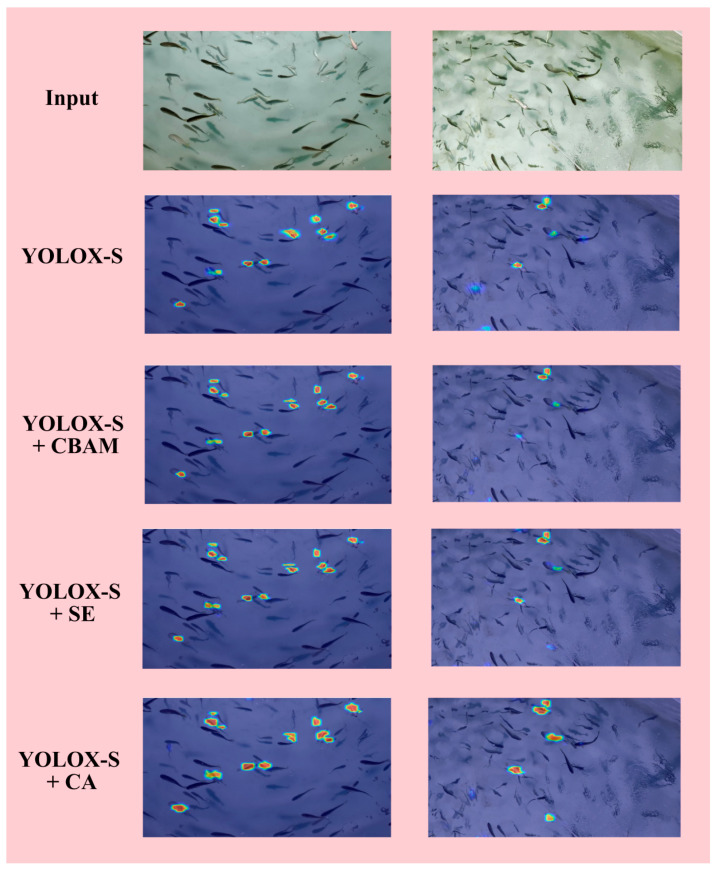
Heat map comparison of the detection performance of YOLOX-S with different attention mechanisms added.

**Figure 8 sensors-23-02835-f008:**
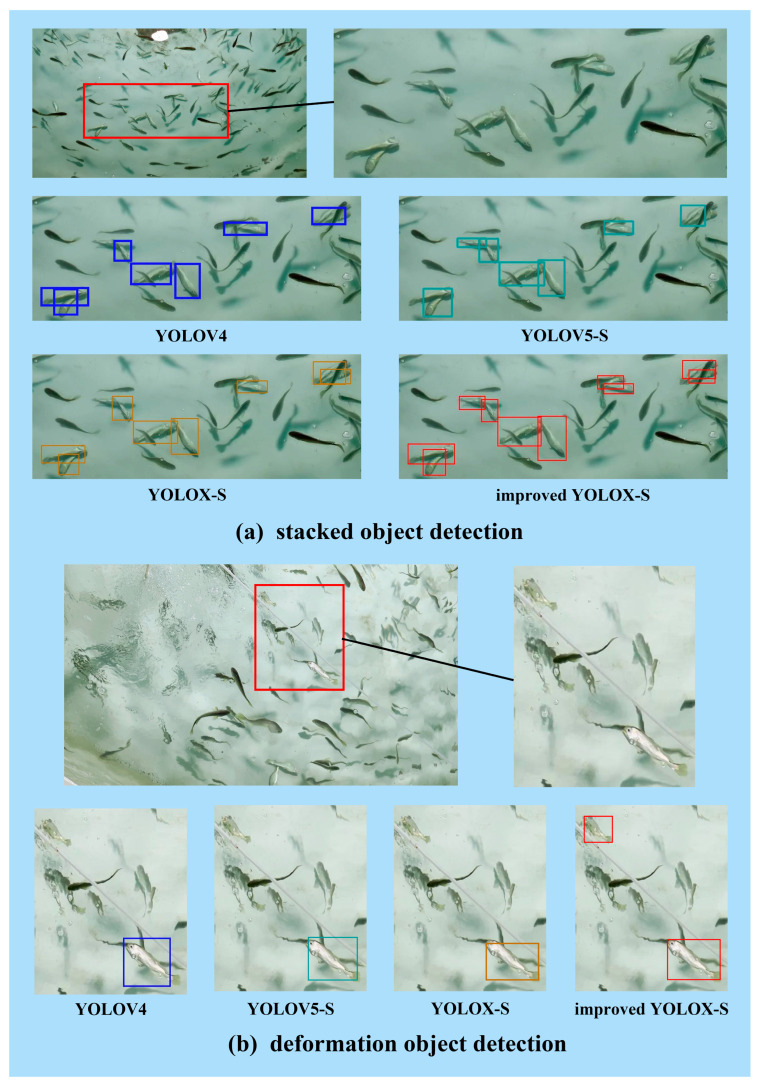
Comparison of the detection performance of different detectors for different problems in the RAS environment: (**a**) stacked object detection; (**b**) deformation object detection; (**c**) occlusion object detection; and (**d**) small object detection.

**Figure 9 sensors-23-02835-f009:**
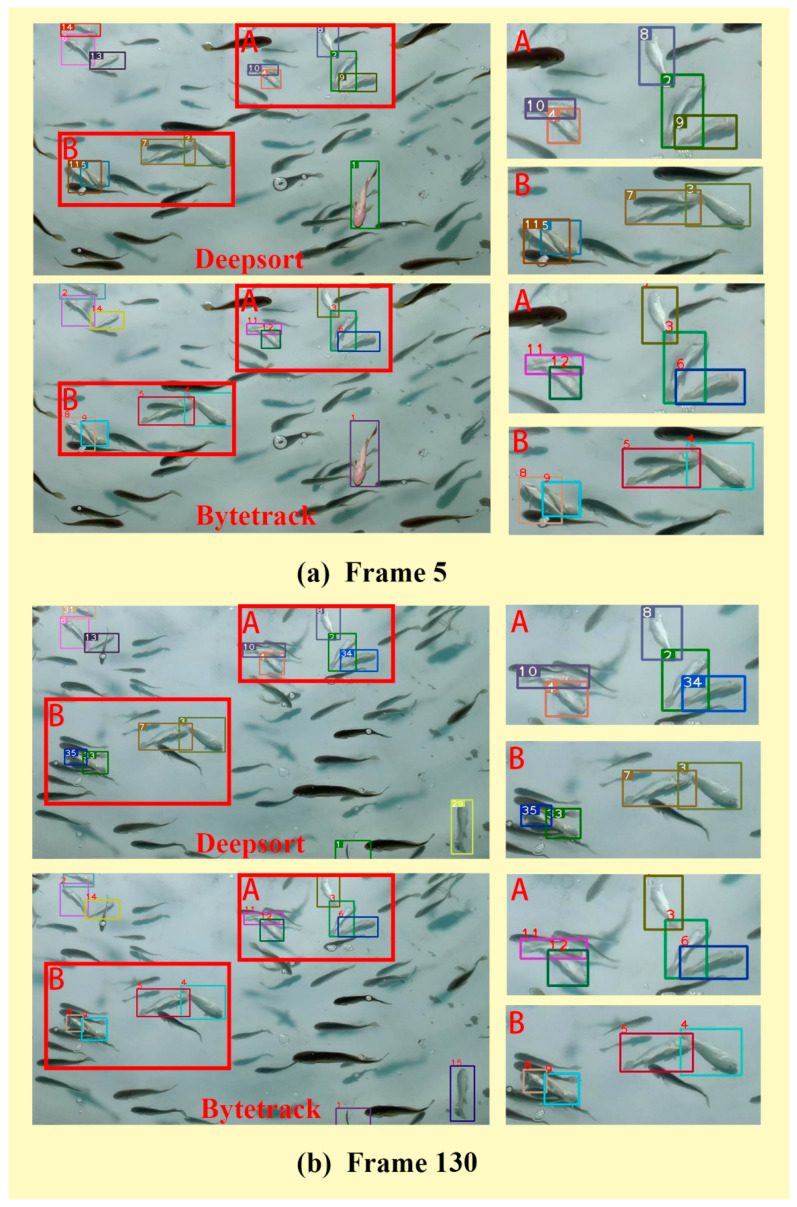
Comparison of the tracking performance of Deepsort and Bytetrack on the abnormal behavior of Larimichthys crocea in a video sequence of the ID changes of different key frames: (**a**) frame 5; (**b**) frame 130; (**c**) frame 150; (**d**) frame 260. Use letters and numbers to highlight the differences between different tracking algorithms. A and B in the figure represent the key focus areas for tracking, and the ID number change of each fish in the area is the main content of the discussion part.

**Figure 10 sensors-23-02835-f010:**
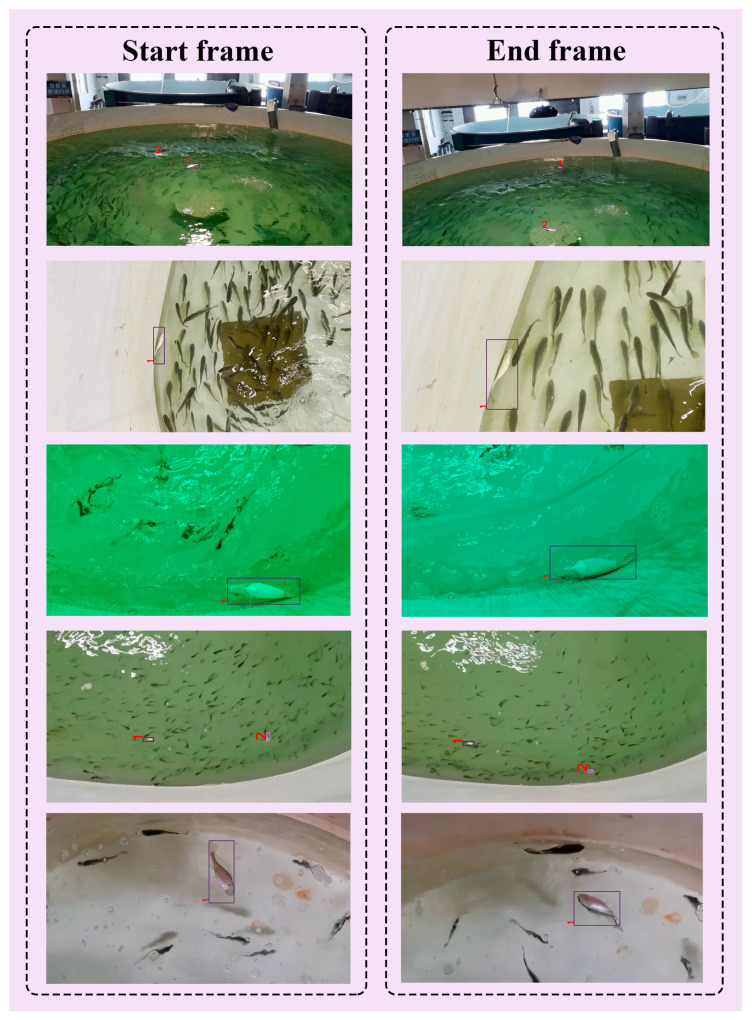
Generalization verification experiment of *Larimichthys crocea* at different growth stages in different cultured fish ponds. The number in the figure represents the ID of the tracked object, which remains stable from the start frame to the end frame.

**Table 1 sensors-23-02835-t001:** Comparison of detection results of different detectors for abnormal behavior.

Methods	F1	Recall	Precision	AP_50_	AP_50:95_	FPS
YOLOV4	0.88	84.32%	91.49%	90.7%	43.7%	44
YOLOV5-S	0.88	87.71%	89.28%	91.5%	50.9%	94.2
YOLOX-S	0.91	89.76%	93.24%	93.3%	54.4%	79.8
Improved YOLOX-S	0.98	97.95%	97.17%	98.4%	70.6%	50.7

**Table 2 sensors-23-02835-t002:** Comparison of tracking results of different video sequences by different tracking algorithms (↑ indicates that a larger value is better, and ↓ indicates that a smaller value is better).

Video Sequence	Video	Tracker	MOTA(↑)	IDF1(↑)	ID Switch (↓)	FPS (↑)
Sequence 1	Appendix A	DeepSort	73.03%	61.80%	56	12.1
	Appendix A	Bytetrack	95.02%	97.70%	1	39.4
Sequence 2	Appendix A	DeepSort	95.90%	69.99%	12	14.8
	Appendix A	Bytetrack	99.40%	96.41%	0	40.2
Sequence 3	Appendix A	DeepSort	97.73%	73.24%	22	12.7
	Appendix A	Bytetrack	99.10%	99.53%	0	39.6

## Data Availability

The data used in this study are available upon request from the authors.

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
