# Peer review of "Abnormal Behavior Monitoring Method of Larimichthys crocea in Recirculating Aquaculture System Based on Computer Vision"

_sensors, 2023, doi:10.3390/s23052835_

Round 1

Reviewer 1 Report

Revised version is well explained and I think good enough for publishing. I accept with this version.

Author Response

We are grateful to you for your careful review and for your recognition of this paper.

Reviewer 2 Report

The paper proposes a method for visual multi-object tracking (MOT) for the Abnormal behavior monitoring method of Larimichthys crocea in recirculating aquaculture. The paper requires significant revision and improvement in the current studies. Furthermore, a major improvement in the description of the results is required. The straightforward use of YOLOX-S can not be termed a significant contribution. 

  1. The experiments are performed on only one type of dataset. Experiments should be performed on some other datasets as well, to validate the proposed method more rigorously.

  2. An analysis could be performed on computational time, to assure the effectiveness of run-time

  3. Many spelling mistakes and poor sentence formation are observed. 

  4. The paper requires new technical contributions. The author proposes a minor variation in the well-known YOLO. 

  5. The figure captions are very brief. This needs to be improved. 

  6. Computational and Mathematical aspects in the current manuscript are missing.  

  7. The performance needs to be extended on varieties of datasets as single data sets do not provide generalization aspects of the algorithm.

Author Response

Thank you very much for your careful review and thoughtful comments on our paper. We understand that the improvement of YOLOX-S is not a significant contribution, and we appreciate your suggestion regarding the improvement of the description of the results. In your comments, we have noticed that you think the literature is not relevant enough, so we have added some relevant references when modifying it. In this paper, we improved YOLOX based on the characteristics of Larimichthys crocea in recirculating aquaculture systems and environmental factors to ensure accurate detection of abnormal behaviors. The tracker Bytetrack was used to track multiple objects with abnormal behavior according to the results of the detector, while previous methods based on deep learning used image features for classification and then tracking were slow and not applicable to fish objects with similar appearance features. We are aware of our shortcomings in research content, but due to time constraints, we have done our best to respond to reviewer's comments. 

Please see the attachment for the details of modification.

Reviewer 3 Report

The paper proposes a monitoring method for the abnormal behavior of Larimichthys crocea in RAS, using an improved YOLOX-s (for detection) and Bytetrack (for tracking). The article compares methods like YOLOV4, YOLOV5-S, and YOLOX-S for detection and DeepSort for tracking versus the proposed method. The supplementary information (videos) helps the reader understand the comparison methods versus the proposed method.

The writing is well structured, and the research topic could interest researchers. 

Minor issues:

-Consider checking the spelling in detail (i.e., equations 1 and 2 (Recall and Precision))

-the google drive link does not work, consider using GitHub or another public repository for additional material (videos)

Author Response

We appreciate the Reviewer’s carefully reviewing and comments, which are very important for us to further improve our work and the quality of the manuscript. The questions raised by reviewer have been carefully revised and checked, and have been answered one by one.

Please see the attachment for the details of modification.

Round 2

Reviewer 2 Report

I appreciate the author's response and understand that in a short time, new data sets cant be created. Overall the revised paper is better than the previous version. I recommend acceptance of the paper.